# Effect of the Shape of Rolling Passes and the Temperature on the Corrosion Protection of the Mg/Al Bimetallic Bars

**DOI:** 10.3390/ma14226926

**Published:** 2021-11-16

**Authors:** Sebastian Mróz, Karina Jagielska-Wiaderek, Piotr Szota, Andrzej Stefanik, Robert Kosturek, Marcin Wachowski

**Affiliations:** 1Faculty of Production Engineering and Materials Technology, Czestochowa University of Technology, 42-201 Częstochowa, Poland; k.jagielska-wiaderek@pcz.pl (K.J.-W.); piotr.szota@pcz.pl (P.S.); andrzej.stefanik@pcz.pl (A.S.); 2Faculty of Mechanical Engineering, Military University of Technology, 00-908 Warsaw, Poland; robert.kosturek@wat.edu.pl (R.K.); marcin.wachowski@wat.edu.pl (M.W.)

**Keywords:** Mg/Al bimetallic bars, explosive welding, groove rolling, microstructure, corrosion resistance

## Abstract

The paper presents the results of experimental tests of the rolling process of Mg/Al bimetallic bars in two systems of classic passes (horizontal oval-circle-horizontal oval-circle variant I) and modified (multi-radial horizontal oval-multi-radial vertical oval-multi-radial horizontal oval-circle-variant II). The feedstock in the form of round bimetallic bars with a diameter of 22 mm and 30% of the outer aluminum layer was made through explosive welding. The bimetallic bars consisted of an AZ31 magnesium core and a 1050A aluminum outer layer. Bars with a diameter of 17 mm were obtained as a result of rolling in four passes. The rolling process in the passes was conducted at two temperatures of 300 and 400 °C. Based on the analysis of the test results, it was found that the use of modified passes and a lower rolling temperature (300 °C) ensures a more homogenous distribution of the plating layer around the circumference of the core and results in an even grain decreasing, which improves the corrosion resistance of bimetallic bars compared to rolling bars in a classic system of passes and at a higher temperature (400 °C).

## 1. Introduction

Products made of magnesium alloys have been drawing more and more attention in many industries for over a dozen years. It relates to the low mass density and high strength of Mg alloys [1,2], while maintaining good plastic properties. As a result, such products are becoming more and more competitive to the widely used steel products. A significant impediment in the wider use of magnesium alloys in technology is their relatively poor corrosion resistance [3,4]. Thus, products made of magnesium alloys are, in many cases, covered with coatings, increasing their corrosion resistance [5,6]. One of many methods of securing products made of magnesium alloys is the application of aluminum as a protective outside layers [7,8]. Aluminum and its alloys exhibit considerably higher corrosion resistance than magnesium does. Thus, it can be expected that by producing a two-layered Mg/Al bar in the process of rolling in elongating has the advantages that both materials, magnesium and aluminum, could be combined. The tightness of the cladding layer (Al) and its appropriate uniform thickness on the perimeter and along the length seem to be the key to the effective inhibition of corrosion of the even more chemically active core material (an Mg alloy). The use of aluminum coatings or layers may be a prospective solution, ensuring an increase in the corrosion resistance of the magnesium core while not causing a significant increase in the weight density of the finished products. The advantage of using coatings of aluminum or its alloys is their excellent corrosion resistance in inert media as well as resistance to mechanical damage or abrasive wear. Due to the spontaneous passivation of aluminum and its alloys, unlike materials based on Fe or Mg, they are highly resistant to corrosion in a common, neutral air environment [9,10,11]. One of the methods ensuring the improvement of corrosion resistance of magnesium alloys is laser surface alloying [7,12] and thermo-chemical treatment [13], which are commonly used techniques for producing layers enriched in Al. The mentioned methods ensure the production of aluminum coatings with a maximum thickness of ~10 μm, which may result in a porous surface and deteriorate corrosion resistance [8]. Thus, it is advisable to search for methods of making aluminum layers of higher thickness, which will effectively protect the magnesium layer. Many works on the production of such layers on multi-layered plates, sheets, and round bimetallic Mg/Al bars can be found in the technical literature. The most commonly used methods include diffusion bonding [14,15], hot pressing [16,17], extrusion [18,19], rolling [20,21] and rolling in grooves [22,23], explosive welding [24,25], forging [26,27], and two-roll casting [28]. Additionally, in this case, some of the mentioned methods do not guarantee the right quality of the joint, the required thickness of the plating layer, and, in addition, in the case of round bimetallic bars, even distribution of the plating layer around the core perimeter [29,30]. The most frequently used methods to produce round bimetallic bars include extrusion [18,19,31] and rolling in grooves [23,30,32]. The feedstock for these processes is made directly in the processes themselves, as in the case of extrusion [18,19] or with the possibility of earlier use of the casting method [22] or explosive welding [30,32]. One of the methods ensuring the right thickness and even distribution on the perimeter of the round bar of the plating layer is the combination of the explosive welding method to produce a bimetallic feedstock, and then rolling in elongating grooves [32,33]. A combination of these methods was used to produce bimetallic bars of steel/Cu [29,34], Al/Cu [30], and, recently, also Mg/Al [22,23,32]. Although the method of explosion welding guarantees obtaining a bimetallic feedstock marked by a high quality of the joint [32], it does not always guarantee obtaining high-quality bimetallic bars while rolling in grooves. Due to the fact that the process of rolling bars in the grooves is marked by a spatial state of deformation and a large inhomogeneity of metal flow in the rolling gap, the distribution of deformations is inhomogeneous [29,34]. Additionally, while rolling bimetallic bars in the grooves, with an unfavorable ratio of the layer thickness to the core diameter, the difference in the plastic flow resistance of individual layers and poorly selected process parameters (deformation, temperature, shape of the grooves) may result in an uneven distribution of the thickness of the plating layer on the perimeter of the bar, and, in extreme cases, delamination of individual components [29,35].

In order to increase the homogeneity of deformation during the rolling of homogeneous bars [36,37,38], and recently also bimetallic bars [33], classic elongating grooves are subject to modification of their shape. In the earlier works of some authors [30,33] it was shown that using multi-radial modified elongating grooves in the circle-oval-circle system to obtain Al/Cu bimetallic bars, such a distribution of strains in individual components of rolled bimetallic bars can be obtained, which will ensure a significant increase in the evenness of the plating layer distribution around the perimeter of the core, with high quality of the joint at the same time. Such grooves were also used for rolling Mg/Al bimetallic bars. In one work [32] it was shown that the distribution of the aluminum layer thickness on the magnesium core was approx. 10% more even compared to Mg/Al bars rolled in the classic circle-oval-circle system. However, available literature data concerning the determination of the effect of process parameters on the pattern of Mg/Al bimetal flow and the possibility of its controlling during groove rolling are very scarce. Thus, the subject matter and scope of the paper will constitute a unique research output that will contribute to the development of a new group of bimetallic products of low specific gravity and enhanced corrosion resistance. The results obtained from the research will provide in the future a basis for carrying out studies within projects of an applied profile. The novelty of this word was to use the explosion welding method for the Mg/Al feedstock production and subsequent groove-rolling process using modified elongating grooves.

That is why the main purpose of using the modified grooves is to increase the uniform distribution of the outer Al layer to increase the corrosion resistance of Mg/Al bimetallic bars. In the existing work, there are no data on the impact of the shape of the grooves and process parameters on the corrosion resistance of Mg/Al bars. Thus, the novelty and the main aim of this work was to determine the impact of the shape of the grooves and the rolling temperature on the corrosion resistance of the bars by using an outer aluminum layer. As part of this research, the rolling process was conducted in two systems of elongating grooves: classic (horizontal oval-circle-horizontal oval-circle variant-I) and modified (multi-radial horizontal oval-multi-radial vertical oval-multi-radial horizontal oval-circle-variant II). The rolling process was conducted in four passes. The obtained Mg/Al bimetallic bars were subjected to complex tests of the layer thickness distribution around the perimeter, structural tests, microhardness, and corrosion resistance tests.

## 2. Materials and Methods

The bimetallic feedstock in the form of Mg/Al round bars used for rolling in grooves was obtained by the explosive welding method. The explosive welding process was conducted in cooperation with the Explomet company (Opole, Poland). The parameters of explosive welding have been described in detail in the previous works of some authors [24,32]. Eight sets of samples were prepared for testing, each consisting of aluminum tubes (grade 1050A) and magnesium bars (grade AZ31), respectively. Chemical composition of the materials used for the tests is given in Table 1. The diameter of the AZ31 bars was 19.2 mm. The outer diameter of the 1050A tubes was 24 mm and the wall thickness of the tube was 1.5 mm. The distance between the magnesium core and the inner diameter of the tube was 0.9 mm. A cylinder system was used for explosive welding [32].

After explosive welding, the obtained Mg/Al bimetallic feedstocks were 400 mm long. Figure 1 shows the view of exemplary Mg/Al bimetallic feedstocks.

The average diameter of the Mg/Al bimetallic feedstocks after explosion welding was 22.5 mm. The obtained bimetallic feedstocks were marked by a slight difference in the thickness of the outer aluminum layer around the perimeter of the magnesium core. The average thickness of the aluminum layer was 1.67 mm and its share in the cross-section of the bimetallic bar was 28% (Figure 2).

One of the deciding factors impacting the even distribution of the plating layer is the shape of the grooves and the distribution of deformations in individual passes. Two systems of elongating grooves were designed for the rolling process: the classic system of grooves, oval-circle-oval-circle (variant I), and its modification, multi-radial horizontal oval-multi-radial vertical oval-multi-radial horizontal oval-circle (variant II). In the case of both variants, the rolling process took place in four passes. The finished round pass was the same in both variants. The rolling feedstock was round bars with 22.5 mm diameter and 100 mm length. After rolling, the obtained round bars had a 17-mm diameter.

Computer simulations with the use of a computer program based on finite element method (FEM) were used to develop the design of the new passes. The assumption of using multiple radii in oval passes was to create the right conditions during deformation, limiting the controlled plastic flow of the magnesium core plating layer and, thus, obtaining a more even distribution of the plating layer around the perimeter of the magnesium core. The deformation pattern for individual variants is presented in Table 2. The shape of two developed groove systems is shown in Figure 3.

Temperature-deformation parameters were selected based on the results of physical and numerical modelings of compression tests of two-layer Mg/Al samples [39,40]. The rolling process was conducted for two temperatures: 300 and 400 °C. After each pass, the samples were reheated to the rolling temperature. The rolling speed was 0.2 m/s. A laboratory two-high rolling mill with a nominal diameter of working rolls of 150 mm was used for the experimental tests (Figure 4). After each pass, templets were collected to determine the thickness distribution of the plating layer around the perimeter of the magnesium core. The bimetallic feedstock was heated in the LAC KC 120/14 (LAC, Židlochovice, Czech Republic) resistance chamber furnace before rolling and before the individual passes.

In order to determine the distribution of the plating layer for each sample collected after a particular pass, 32 measurements were conducted, which corresponded to the multiple of the angle of 11.25°, starting from the orientation (vertical symmetry axis of the groove) in the N direction. The evenness of the thickness distribution of the 1050A layer on the core of the AZ31 bar was determined using the coefficient of unevenness of the thickness distribution of the *K_plat_* plating layer [40], defined as the ratio of the maximum thickness of the plating layer (*t*_max_) to the minimum thickness (*t*_min_), in the cross-section of the finished Mg/Al bimetallic bars, which can be determined using the following Equation (1):(1)$$K_{plat}=\frac{t_{\max}}{t_{\min}}$$

A detailed description of the methodology used to determine the analyzed coefficient *K_plat_* is presented in [40,41]. In order to ensure the correctness of the tests, the aluminum layer thickness distribution was determined for three samples from batch bars and for three samples taken from corresponding rolled bars for the analyzed variants. The obtained difference in measurements for the charge after the explosion welding process did not exceed 5%, and for the rolled bars it was slightly higher, amounting to 8.6%. The increase in unevenness was closely related to the rolling operation and can be minimized by modifying the rolling equipment. This confirmed the correctness and repeatability of both the charge production processes and the finished bars used.

Microstructural tests of Mg/Al bimetallic bars included observations on an optical microscope Olympus LEXT OLS 4100 (Olympus, Tokyo, Japan) and a scanning microscope JEOL JSM-6610 (JEOL, Tokyo, Japan). The samples were included in the resin, sanded with 80, 320, 500, 800, 1200, 2400, and 4000 gradations, and then polished with a diamond paste of 3 mm and 1 mm. In order to reveal the microstructure of the AZ31 alloy, samples were etched with a reagent composed of 19 mL of ethanol, 2 mL of acetic acid, and 1 g of picric acid (digestion time of 30 s). Revealing the microstructure of 1050A was carried out using 1% HF with an etching time of 60 s.

To assess the impact of the groove shape and the rolling temperature on the corrosion resistance of Mg/Al bimetallic bars, potentiodynamic polarization curves were conducted in a 0.5M Na_2_SO_4_ solution acidified to pH = 4.0. The electrodes in the form of rotating disks were used for electrochemical tests, in which fragments of the side surfaces of the tested samples with an area of 0.2 cm^2^ were used for the electrodes. All potentiodynamic tests were performed at the temperature of 25 ± 0.1 °C, with the rotational speed of the disc equal to 12 rpm^−1^ and the scanning speed of the potential of 0.005 V∙s^−1^ using its shift from *E*_0_ value of 0.3 V lower than *E*_corr_ to *E*_1_ = +0.5 V (regarding AgCl/Ag). This type of methodology ensured the production of repeatable passive layers (in the cathode range, the natural, spontaneously formed oxide layers were reduced during surface preparation), and, on the other hand, it limited material consumption, thanks to fast scanning and turning off polarization at relatively low potential values. Each time before drawing the potentiodynamic curve, the test sample was held for 15 min in the corrosive solution, i.e., until the corrosion potential reached a stationary value.

## 3. Results and Discussion

The collected templets after individual passes were analyzed in terms of changes in shape and dimensions on the cross-section. Figure 5 shows that each pass, regardless of the used variant, was properly filled with a bimetallic flow. In any case, there was no overflow of the pass and no delamination was observed at the joint of the components. Rolling at temperature of 300 °C caused that the difference in the plastic flow resistance of the Al layer and the magnesium core decreased, which reduced the uncontrolled flow of the soft aluminum layer from the magnesium core [26]. As a result, the greater volume of the deformed flow moved more intensively in the longitudinal direction at the expense of reducing its width (widening). Thus, the width of the finished bars rolled at the temperature of 300 °C was smaller (Figure 5a,c) compared to the width of the bars rolled at the temperature of 400 °C (Figure 5b,d). The lower rolling temperature enabled us to obtain a greater coefficient in relation to the assumed rolling pattern (Table 2), which amounted to 1.85 elongation in variant II.

Figure 6 shows the average thickness of the plating layer for the feedstock after explosive welding and the finished Mg/Al bars after rolling with the use of two types of grooves.

The data presented in Figure 6 show that the average thickness of the plating layer varied depending on the rolling variant and the rolling temperature. According to the data in Figure 6, the lower rolling temperature resulted in a smaller spreading and greater elongation of the Mg/Al band. It influenced the limitation of the uncontrolled flow down of the Al layer in individual passes, thanks to which the average thickness of the plating layer was greater compared to bars rolled at the temperature of 400 °C, where we can observe higher flowing down of the cladding layer. The greater thickness of the plating layer may increase the corrosion resistance of the Mg/Al bars. The average thickness of the plating layer for the bars rolled at the temperature of 300 °C was higher by approx. 5% in relation to bars rolled at the temperature of 400 °C. The impact of the applied system of passes was much smaller in relation to the rolling temperature. For the system of modified grooves, regardless of the rolling temperature, the thickness of the plating layer was slightly lower than the thickness of the Al layer obtained for the system of classic passes. This difference was mainly due to the greater elongation of the bars rolled in the system of modified passes.

The main idea of using modified grooves (variant II) was to limit the uncontrolled plastic flow of the plating layer, thanks to which it will be possible to obtain Mg/Al bars with a more even distribution of the plating layer around the perimeter of the magnesium core. Limiting the local thinning of the plating layer on the perimeter and obtaining its even distribution is an important factor impacting the corrosion resistance of Mg/Al bars. Figure 7 presents the results of the calculations of *K_plat_* coefficient calculated for finished bars rolled in two systems of passes.

The data presented in Figure 7 show that the developed new system of modified passes (variant II) had a positive effect on the thickness distribution of the Al layer. In each of the analyzed rolling variants, there was an increase in the unevenness of the plating layer distribution in relation to the feedstock after explosive welding; however, during rolling at the temperature of 300 °C with the use of modified grooves, the most even plating thickness distribution was obtained. For explosive welding, components in the form of AZ31 bars and Al tubes with perfect geometry were used. Thus, *K_plat_* coefficient, marked by the uneven distribution of the plating layer, was close to 1 (*K_plat_* = 1.12). Due to the fact that rolling in passes is marked by a spatial state of deformation and the lack of possibility to fully control the flow of the bimetallic band, it is natural that *K_plat_* for bimetallic bars, after rolling in grooves, will increase in relation to the bimetallic feedstock. The modification of the shape of the passes, consisting of replacing the single-radius surface with multi-radial surfaces with straight sections, had a positive effect on the change in the deformation distribution and the evenness of the plastic flow of individual components in the Mg/Al bimetallic bar during rolling in the rolling gap (variant II). The new shape of the grooves enabled us to limit the accumulation of the plating layer material at the ends of the grooves towards the horizontal axis of the pattern symmetry and to reduce the thinning of the layer towards the bending at the same time. Thus, regardless of the rolling temperature, *K_plat_* coefficient increased to a smaller extent in variant II in relation to Mg/Al bimetallic bars rolled in the classic system of passes (variant I). Rolling bars at the temperature of 300 °C, regardless of the applied rolling variant, had a positive effect on the evenness of deformation and plastic flow of the bimetal components. Thus, for Mg/Al bars rolled at 300 °C, *K_plat_* coefficient was lower compared to the bars rolled at 400 °C. The most even distribution of the plating layer was obtained for Mg/Al bars rolled in passes modified for the rolling temperature of 300 °C. Detailed changes of *K_plat_* coefficient for individual rolling variants are presented in Table 3.

The data analysis in Table 3 shows that the rolling of bars in the classic system of passes (variant I) increases the uneven distribution of the plating layer compared to the bimetallic feedstock by approx. 20% (temperature 300 °C) and by approx. 30% (temperature 400 °C). The introduction of modified passes (variant II) significantly reduced the unevenness of the Al layer distribution on the perimeter of the Mg/Al bar. For the bars rolled at 300 °C, increasing *K_plat_* coefficient in relation to the bimetallic feedstock totaled below 10%, and for the temperature of 400 °C, only 15%. Both values obtained for the rolled bars according to variant II, regardless of the rolling temperature used, were lower than the values obtained for bars rolled in the classic system of grooves (variant I). Comparing the obtained *K_plat_* coefficients for individual variants, a greater evenness of the plating layer can be found for modified grooves, a reduction of *K_plat_* by 8% for 300 °C and 11% for 400 °C, respectively. 

Figure 8 and Figure 9 show the results of microstructural tests for bimetallic bars rolled in classic and modified passes at the temperature of 300 °C and 400 °C, (cross-sections of samples).

Based on the analysis of the microstructure, it can be concluded that bimetallic bars rolled at a lower temperature (Figure 8a,b) are marked by the presence of the finest grain of the AZ31 alloy in the joint area of approx. 10–15 µm. Despite the similar grain size, the joint in the sample taken from the bar rolled in the modified passes was free from imperfections, unlike the sample taken from the bar rolled in the classic passes, in which local delamination and cracks of the AZ31 alloy in the joint area were found (the area marked in red). Samples distorted at a higher temperature (Figure 8c,d) were characterized by both a larger grain size of the AZ31 alloy and the presence of a diffusion zone at the joint line with a visible presence of numerous imperfections in the form of cracks and delaminations (red arrows). In the case of a sample taken from the bar rolled in classic passes (Figure 8c), the differences were the least visible, and the microstructure itself was similar to the sample shown in Figure 8a with the presence of a larger grain in the joint area (15–20 µm). In the case of a sample taken from the bar rolled in classic passes (Figure 8d), inhomogeneities in the granular structure were found, a fine-grained microstructure (approx. 10–15 µm) was noticed directly at the joint line, and there was an area with very large grains (40–50 µm) with a noticeable presence of twin boundaries (the area marked in red) behind this zone.

The microstructural analysis of the 1050A outer layers for most of the tested samples enabled the identification of large inhomogeneities in the grain sizes, regardless of the applied process parameters. The grain size ranged from 20 µm up to 200 µm. Most of the samples showed the dominance of the coarse-grained microstructure. The sample taken from the bar rolled in passes modified at the temperature of 300 °C (Figure 9b), in which the microstructure was much more distorted because of plastic working, clearly differed from this trend. The most damaged part of the material passes modified at the temperature of 300 °C (Figure 9b) was the central part of the 1050A alloy layer, where there were areas characterized by a relatively small grain size (5.0 ÷ 31.4 µm). 

The implemented modification of the passes, consisting of substituting the single-radial pass surface with multi-radial surfaces with straight line segments, had the favorable effect of changing the strain, stress distribution, and the plastic flow of individual components in the Mg/Al bimetallic bar in the roll gap during rolling. It affected the higher decrease of the grain size. The modified roll pass design helped to limit the build-up of the cladding layer material and the ends of the passes in the direction of the axis of symmetry of the pass and, at the same time, to reduce the layer thinning in the rolling reduction direction. The new pass shape of passes reduced the non-uniform distribution of the deformation and stress value in the rolling direction. Additionally, lowering the rolling temperature to 300 °C had a favorable effect on the homogeneity of the deformation and plastic flow of the bimetal components and, consequently, the decreasing of the grain size. The most uniform distribution of the cladding layer and lower grain size was obtained for the modified passes for the rolling temperature of 300 °C.

More homogenous deformation in individual passes for the system of modified grooves and a lower rolling temperature resulted in a more homogenous grain size for the analyzed sample.

To determine the diffusion zone, the EDS analysis was performed for samples taken from rolled bars in two variants and at two temperatures, including a map of alloying elements on the AZ31–1050A joint surface (Figure 10). Microstructural observations carried out on a scanning electron microscope enabled us to identify the diffusion zone in individual samples. In the case of Al-Mg bars distorted at a temperature of 300 °C, a relatively small diffusion zone with a thickness of approx. 1–1.5 µm was observed, while the thickness of the layer in the sample obtained from the bar rolled in the modified passes (variant II) was slightly lower; however, due to the inaccuracy of the EDS method, it could not be confirmed at this stage of the research. As predicted, samples distorted at higher temperature (400 °C) were characterized by much thicker diffusion layers, which enabled their more detailed analysis. For each of the analyzed samples, the diffusion zone consisted of two layers with different shares of magnesium and aluminum. Earlier studies conducted by the authors on the deformation at high temperature of the Al-Mg joint indicated that the layers of the diffusion zone consisted of intermetallic phases, respectively: β (Mg_2_Al_3_) on the aluminum side and γ (Mg_17_Al_12_) on the magnesium side [42]. In the case of a sample taken from bars rolled in classic passes, the diffusion zone was approx. 13 µm thick with participation of individual phases of 10 µm Mg_2_Al_3_ and 3 μm Mg_17_Al_12_. In the sample taken from bars rolled in modified passes, the zone was slightly smaller and was characterized by a thickness of approx. 11 µm with the presence of 7 µm Mg_2_Al_3_ and 4 μm Mg_17_Al_12_.

The obtained polarization curves were drawn for material in their initial state (AZ31 alloy bar) and for Mg/Al bimetals rolled in two systems of elongating passes: classic (variant I) and modified (variant II) are shown in Figure 11. The feedstock materials in the form of AZ31 alloy bars and 1050A aluminum tubes used to produce bimetal, despite the small distance in the voltage series of metals (the normal potential for Mg is −2.37 V and for Al −1.66 V), showed definitely different corrosion parameters in an acidic environment [43,44]. As can be seen from the diagrams presented in Figure 11, in the applied corrosive environment, aluminum achieved anode currents more than three times the values lower than the magnesium alloy, which was the bimetal core. For the AZ31 alloy in corrosive solution with pH = 4.0, anode currents (*i*_a_) reached values of approx. 270 mA/cm^2^, while they did not exceed 1.4·10^−2^ mA/cm^2^ for aluminum. As it is known, in solutions with a pH of range from 4 to 8, in contrast to magnesium and its alloys, a permanent hydroxide was formed on the surface of aluminum, which was a passive layer, effectively protecting the surface of this material against corrosion [45]. 

Comparing the schemes of the potentiodynamic curves of bimetals after rolling in the grooves, presented in Figure 11, it can be seen that their schemes were impacted by both the rolling method and the rolling temperature. As can be seen, the bimetals, after rolling in classic grooves (variant I), were marked by the lowest values of the corrosion potential (*E*_corr_) and, thus, the earliest etching of the surface. In addition, the highest values of anode currents (*i*_a_) were also found for this type of rolling. Modification of the rolling process (variant II) caused a significant shift of *E*_corr_ towards more positive values. Such an increase of *E*_corr_ was observed for samples after rolling according to variant II (system of modified passes), regardless of the temperature used in this process, and it proved that the processes of active etching of the surface were delayed. Precise values of *E*_corr_ and *i*_a_ read at *E* = 0.25 V are presented in Table 4.

To determine the corrosion rate of Mg/Al bimetals, based on the drawn polarization curves, the polarization resistance (*R*_p_), which is a parameter that determines the corrosion rate, was calculated. Figure 12 presents the relationships of linear polarization Δ*E* =*E* − *E*_corr_ = f (Δ*i*) for equal potentials *E*_corr_ ± 20 mV for bimetals after various rolling variants. For comparison, Figure 12 also present the same relationships for the starting materials from which the tested bimetals were made. As it is known, for potentials slightly different from the corrosion potential (*E*_corr_ = ±20 mV), i.e., in the range in which the Stern–Hoar equation [9] is fulfilled, the external current density is a linear function of the potential, and the slope of the corresponding lines is a measure of the polarization resistance (*R*_p_) inversely proportional to the corrosion rate (*i*_corr_). Table 4 shows the characteristic values determined from the polarization curves that described the corrosion properties of the tested bimetal, i.e., the corrosion potential (*E*_corr_), anode current (*i*_a_), and corrosion current density (*i*_corr_), determined based on the value of the polarization resistance (*R*_p_) [46].

As can be seen from the data in Table 4, Mg/Al bimetals, after rolling processes, showed slightly more favorable corrosion parameters (lower values of *i*_corr_ and *i*_a_) than Al, from which the coating was made. This slight improvement in the corrosion parameters of aluminum was probably due to the reduction of the grain size of the material due to the fragmentation of the primary grains during the deformation of the bars in individual passes. Similar results of an increase in the corrosion resistance of aluminum as a result of grain refinement were obtained after the forging (RS) [47] and angular pressing (ECAP) [48] processes. The authors attributed the increase in the corrosion resistance of the material to the formation of a compact and tight passive layer, the formation of which was fostered by both the high density of grain boundaries and dislocations. Detailed considerations on the influence of grain refinement on the corrosion resistance of aluminum are presented in [49]. While confirming the beneficial effect of grain grinding on the corrosion resistance of Al, the authors point out that the intensity of this influence depends on the corrosive environment. They also note that both the increase in the dislocation density and the associated stresses as well as the texture may affect the corrosion rate.

From a corrosive point of view, due to the slowest corrosion processes (low values of *i*_corr_), the samples rolled at 300 °C are worthy of noticing. Regardless of the applied rolling variant, these bimetals corroded at the same rate, of 0.3·10^−3^ mA/cm^2^, while after rolling at 400 °C they corroded slightly faster (*i*_corr_ = 1.1·10^−3^ mA/cm^2^). However, while the corrosion rates of the bimetals rolled at lower temperatures were the same, it should be noted that the more favorable values of both the corrosion potential and the anode currents were achieved by rolling with the modified method. The best corrosion characteristics obtained for Mg/Al bars rolled in modified passes for the rolling temperature of 300 °C probably reflect the fine-grained structure [49,50] and a more even distribution of the plating layer (Figure 7).

## 4. Conclusions

The use of modified passes in the circle-oval-circle system for rolling Mg/Al bimetallic bars results in the production of bars with a more even distribution of the plating layer around the core perimeter. A properly selected rolling temperature (300 °C) in combination with the applied deformation in the particular passes and the modification contributes to reaching bimetallic bars with high geometric accuracy of the cross-section. A more even deformation in the Al plating layer resulting from the deformation method and the reduced temperature results in an even grain fragmentation, which improves the corrosion resistance of bimetallic bars.

Rolling of Mg/Al bimetallic bars in modified passes at a temperature of 300 °C causes the smallest grain size of aluminum, which results in higher corrosion resistance of the final bars. In addition, it was found that bars rolled in modified passes are characterized by a slightly thinner diffusion zone and a lower presence of cracks and delamination in the joint area compared to bars rolled in a classic pass system, regardless of the rolling temperature.

The corrosion resistance of Mg/Al bimetallic bars is strictly dependent on both the temperature and the applied rolling method (the shape of the grooves). Conducting the rolling processes at the temperature of 300 °C slows down the etching processes of the bimetal surfaces, effectively reducing their corrosion rate. Moreover, the corrosion resistance of the Mg/Al bars is closely related to the thickness and uniform distribution of the plating layer, defined as the *K_plat_* coefficient. The increase in the uniform distribution of the outer layer of Al as a result of the use of modified blanks amounted to approx. 10%, which had a direct impact on the general corrosion resistance of the entire Mg/Al system.

Among the applied rolling variants, bimetals rolled in modified passes are marked by better corrosion parameters, which is reflected in the shift of the corrosion potential towards the positive and lower anode currents. Higher *E*_corr_ values of the samples after modified rolling may prove that this treatment (compared to classic rolling) may additionally delay the start of etching of the bimetal surface in a corrosive environment.

The use of multi-radial modified elongating passes for rolling Mg/Al bimetallic bars, which were influenced by the plastic flow of the bimetal components and ultimately resulted in the finished bars with a uniform cladding layer distribution on the core, resulted in small dimensional deviations. Moreover, the determination of the effect of cladding layer thickness distribution and the cladding layer thickness on the magnesium core on the corrosion resistance of Mg/Al bimetallic bars has not been the subject of research so far.

## Figures and Tables

**Figure 1 materials-14-06926-f001:**
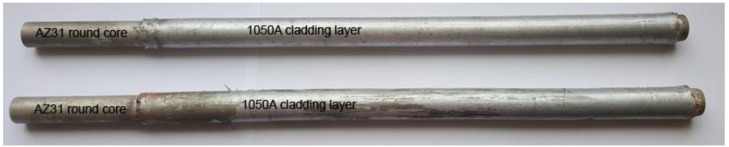
View of round Mg/Al bimetallic feedstocks after explosive welding.

**Figure 2 materials-14-06926-f002:**
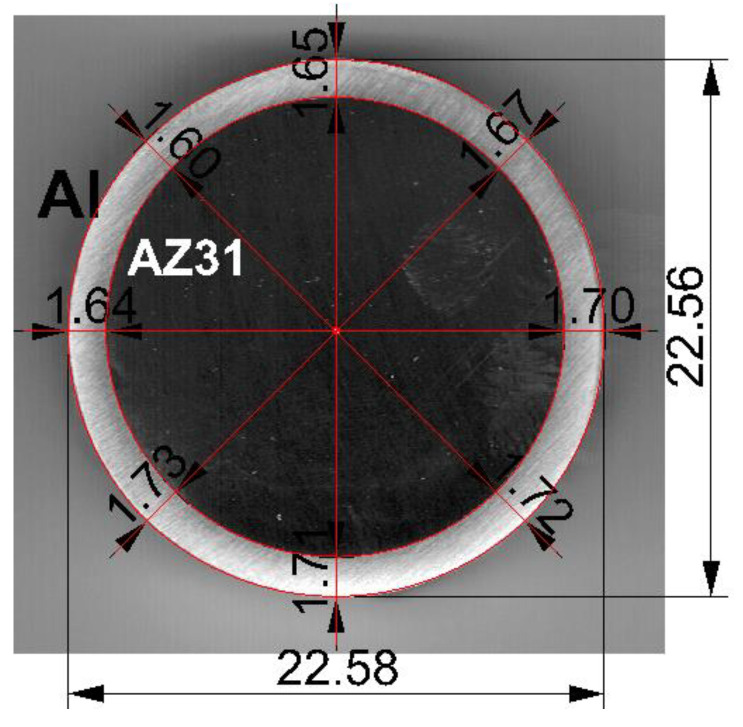
An exemplary shape of the Mg/Al bimetallic feedstock (cross-section) after explosive welding.

**Figure 3 materials-14-06926-f003:**
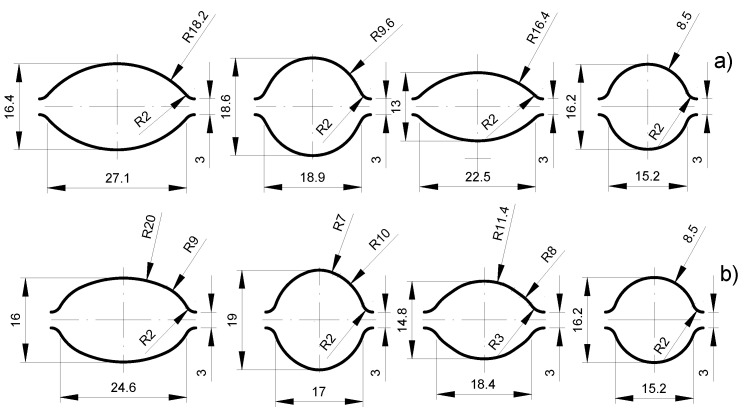
The shape and dimensions of the designed elongating passes: (**a**) classic system (variant I), (**b**) multi-radial modified system (variant II).

**Figure 4 materials-14-06926-f004:**
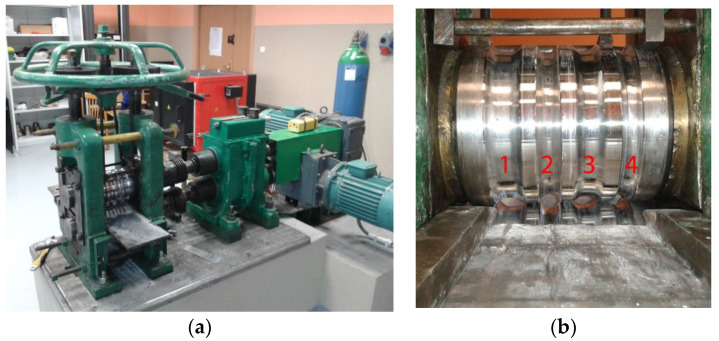
Laboratory rolling mill D150 mm, general view (**a**); arrangement of the grooves along the width of the roll (**b**).

**Figure 5 materials-14-06926-f005:**
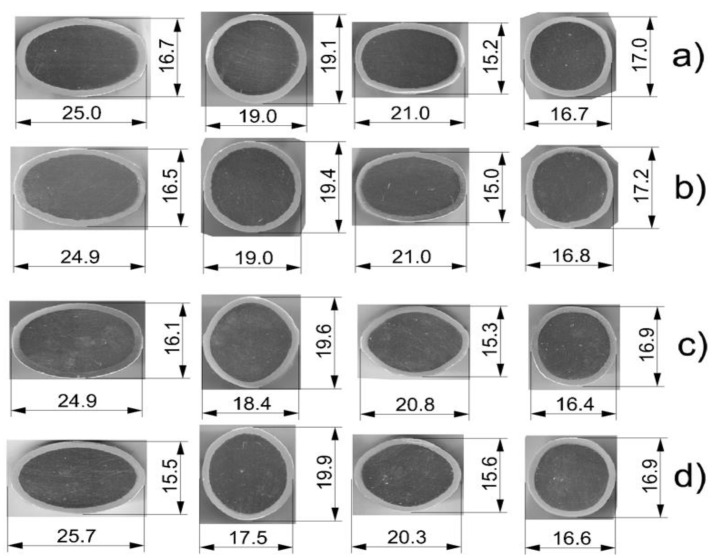
The shape and dimensions of the templets after rolling: (**a**) classic system of passes (variant I), rolling temperature of 300 °C; (**b**) classic system of passes (variant I), rolling temperature of 400 °C; (**c**) modified system of passes (variant II), temperature of 300 °C; (**d**) modified system of passes (variant II), temperature of 400 °C.

**Figure 6 materials-14-06926-f006:**
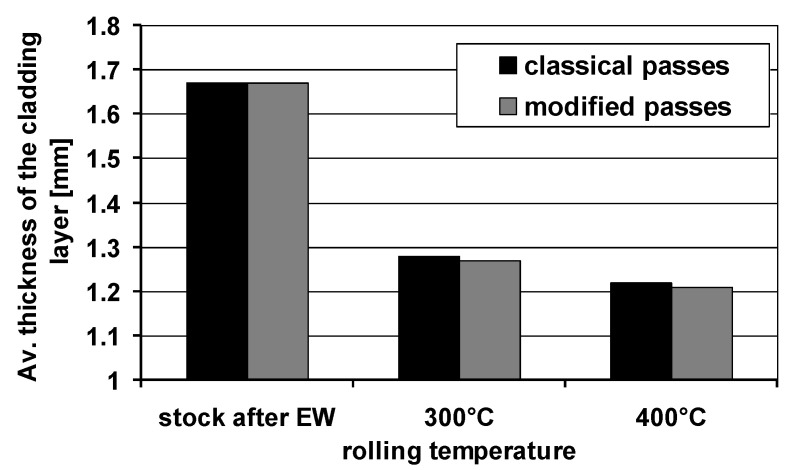
Average thickness of the plating layer around the perimeter of the magnesium core.

**Figure 7 materials-14-06926-f007:**
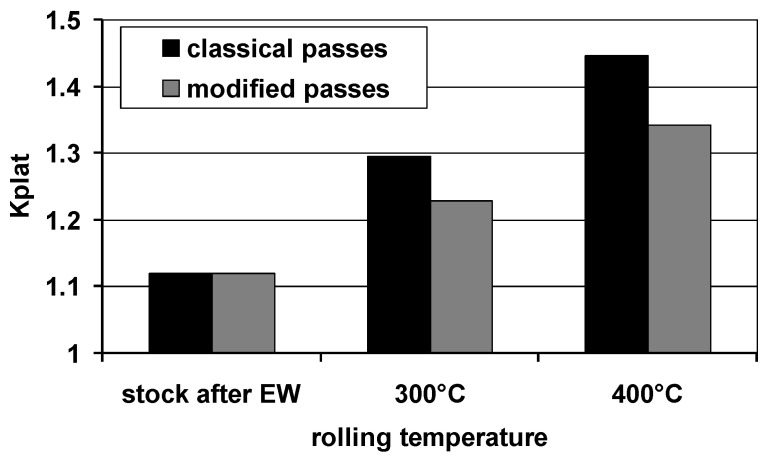
Values of *K_plat_* coefficient depending on the rolling conditions.

**Figure 8 materials-14-06926-f008:**
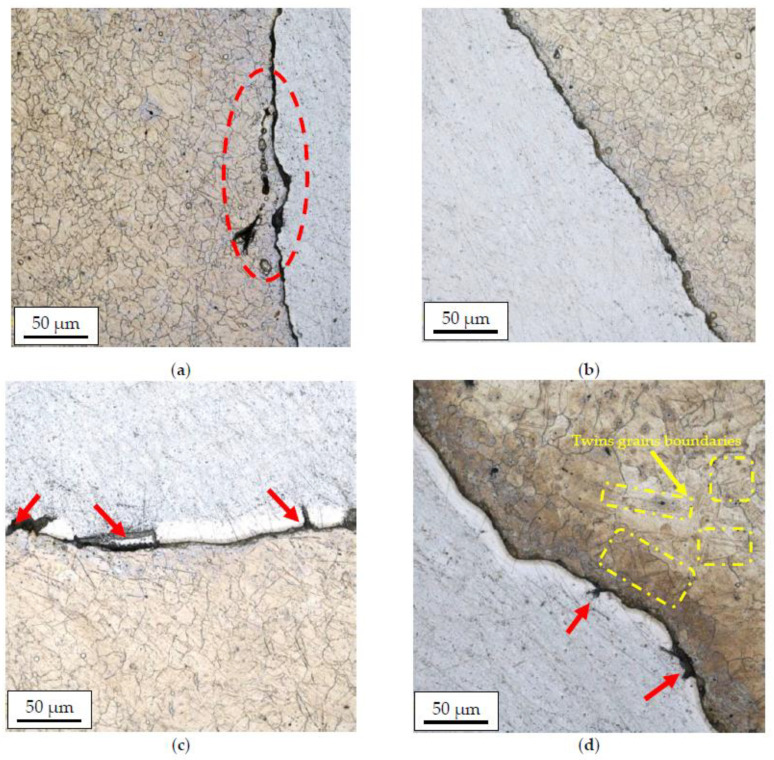
The AZ31 alloy microstructure after rolling in classic passes (variant I), temperature of 300 °C (**a**); modified passes (variant II), temperature of 300 °C (**b**); classic passes (variant I), temperature of 400 °C (**c**); modified passes (variant II), temperature of 400 °C (**d**).

**Figure 9 materials-14-06926-f009:**
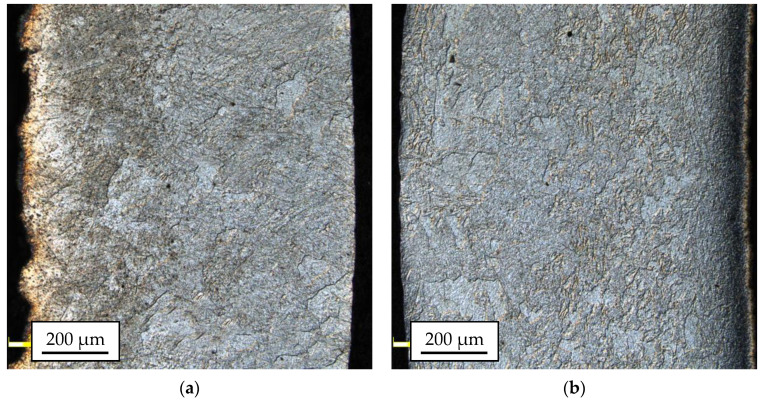
The 1050A microstructure after rolling in classic passes (variant I), temperature of 300 °C (**a**); modified passes (variant II), temperature of 300 °C (**b**); classic passes (variant I), temperature of 400 °C (**c**); modified passes (variant II), temperature of 400 °C (**d**).

**Figure 10 materials-14-06926-f010:**
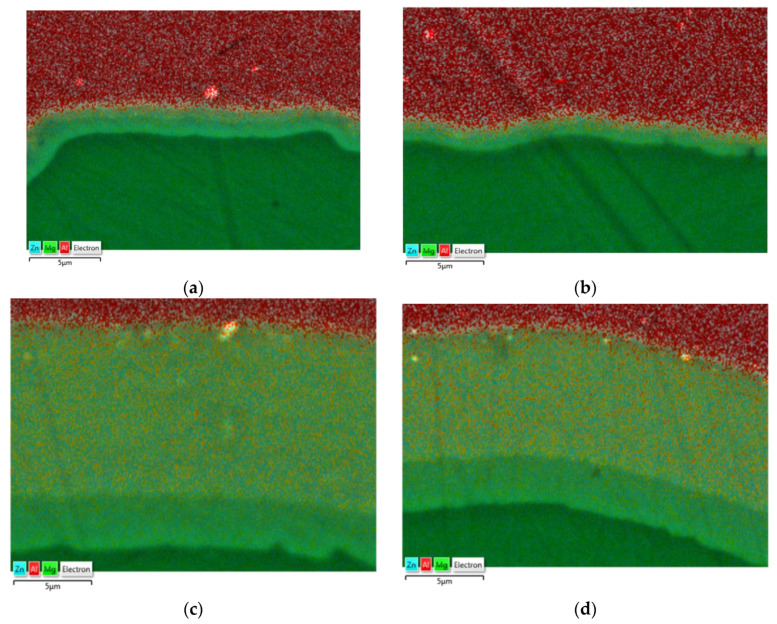
The EDS analysis of the joint zone: classic passes (variant I), temperature of 300 °C (**a**); modified passes (variant II), temperature of 300 °C (**b**); classic passes (variant I), temperature of 400 °C (**c**); modified passes (variant II), temperature of 400 °C (**d**).

**Figure 11 materials-14-06926-f011:**
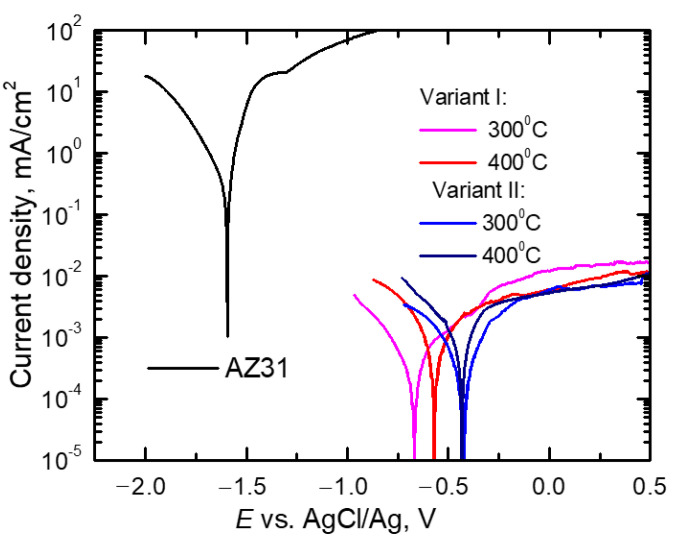
Potentiodynamic polarization curves for AZ31 and Mg/Al bimetals rolled at 300 °C or 400 °C for the classic (variant I) and modified system of grooves (variant II).

**Figure 12 materials-14-06926-f012:**
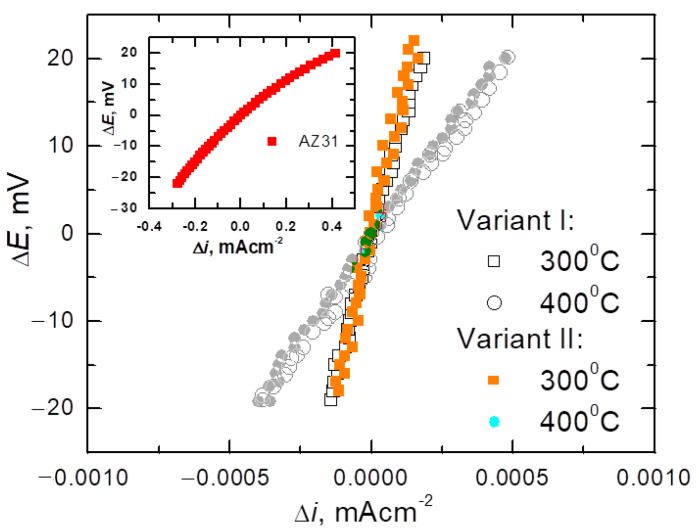
Measurements of linear polarization Δ*E* = *E* − *E*_corr_ = f(i_outside_) of input material (AZ31) and bimetals after classic (variant I) or modified (variant II) rolling at the temperature of 300 °C or 400 °C.

**Table 1 materials-14-06926-t001:** Chemical composition of the materials used for the tests [32] (reprinted with kind permission of Springer).

Material	Chemical Composition, % Mass.								
AZ31	Mg	Mn	Cu	Zn	Ca	Al	Si	Fe	Ni
	ball.	0.24	–	0.72	–	2.8	0.01	0.003	0.001
1050A	Al	Fe	Cu	Mn	Mg	Zn	Ti	Si	Pb
	ball.	0.18	0.002	0.003	0.002	0.008	0.020	0.06	–

**Table 2 materials-14-06926-t002:** Pattern of deformations used in the rolling process of Mg/Al bimetallic bars.

Rolling Variant	Coefficient of Elongation				Av. Coefficient of Elongation	Total Elongation
	Pass no. 1	Pass no. 2	Pass no. 3	Pass no. 4		
I	1.20	1.20	1.10	1.10	1.15	1.75
II	1.25	1.15	1.13	1.08	1.15	1.75

**Table 3 materials-14-06926-t003:** Change of *K_plat_* coefficient in Mg/Al bars after rolling in grooves.

Temperature °C	Feedstock *K_plat_*	Variant I *K_plat_*	Change, % 2/3	Variant II, *K_plat_*	Change, % 2/5	Change, % 3/5
1	2	3	4	5	6	7
300	1.12	1.34	19.6	1.23	9.8	−8.2
400	1.12	1.45	29.5	1.29	15.2	−11.0

**Table 4 materials-14-06926-t004:** Parameters determining the corrosion resistance of the initial materials and the Mg/Al bimetal after various rolling variants.

Material/ Rolling Variant	Rolling Temperature [°C]	*E*_corr_, [V]	*i*_a_ [mA/cm^2^]	*R*_p_ [Ω·cm^2^]	*i*_corr_ [mA/cm^2^]
AZ31	-	−1.6	270	60	860·10^−3^
variant I	300	−0.66	1.4·10^−2^	160·10 ^3^	0.3·10^−3^
	400	−0.57	1.0·10^−2^	50·10 ^3^	1.1·10^−3^
variant II	300	−0.43	0.7·10^−2^	160·10 ^3^	0.3·10^−3^
	400	−0.43	0.7·10^−2^	50·10 ^3^	1.1·10^−3^

## Data Availability

The data presented in this study are available on request from the corresponding author.

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
