# Peer review of "Effect of the Shape of Rolling Passes and the Temperature on the Corrosion Protection of the Mg/Al Bimetallic Bars"

_materials, 2021, doi:10.3390/ma14226926_

Round 1
Reviewer 1 Report
The author studied the distribution of the plating layer and corrosion protection of the AZ31/1050A bimetallic bars alloy with 300/400℃ after the different shape rolling process. The rolling paths were well designed. However, the experimental methods were too simple and the paper was lack of analysis in depth. Therefore, I have to refuse this manuscript and the following suggestions can be found for their reference:
- The author has paid too little attention to the microstructure analysis only through OM. The grains size of different part should be measured precisely according to the main standard. This is strongly advised to analyze by EBSD.
- The Fig.9 is not clear enough to support the point of view.
- The result from Fig.11(a) is not necessary and the author devoted a whole paragraph in the paper to explaining this well-known result.
- Almost no analysis is given by author to explain the difference of the corrosion protection with two different rolling paths and temperature,especially the effect by the thickness or unevenness of the diffusion layer and the phase change.
- The author claims that the ia of AZ31 alloy reach values of approx. 350 mAcm-2 in line 344. However, the value is 270 from Table 4. Some other comments have the same problems. Please check the data.
- The writing of unit should be standardized.The ‘mAcm-2’ in Fig.12 and in whole paper should be uniform with the Table 4.
- The reason why and how the microstructure is influenced by the shape of the passes should be further discussed.
Author Response
REVIEWER #1
The additional text was marked as yellow.
- The author has paid too little attention to the microstructure analysis only through OM. The grains size of different part should be measured precisely according to the main standard. This is strongly advised to analyze by EBSD.
The authors fully agree with the comment that average grains size should be measure more precisely. In this case, the authors decided to measure the average grain size of the A1050 and AZ31 basing on light microscopy images of randomly chosen areas on the samples. Grain size calculation was carried out using a DigitalSurf MountainsMap computer image analyzer. Proper information about the calculation method and results is added to the manuscript. EBSD is recommended to get a more valuable microstructure description but unfortunately, the authors don’t have an EBSD detector but in future research, the authors will try to get access to EBSD by other research units.
- The Fig.9 is not clear enough to support the point of view.
Fig. 9 was changed
|
|
||||
|
(a) |
(b) |
||||
|
|
||||
|
(c) |
(d) |
Figure 9. The 1050A microstructure after rolling in classic passes (variant I), temperature of 300°C (a); modified passes (variant II), temperature of 300°C (b); classic passes (variant I), temperature of 400°C (c); modified passes (variant II), temperature of 400°C (d).
- The result from Fig.11(a) is not necessary and the author devoted a whole paragraph in the paper to explaining this well-known result.
As a reviewer suggestion Fig.11(a) and a whole paragraph were deleted
- Almost no analysis is given by author to explain the difference of the corrosion protection with two different rolling paths and temperature, especially the effect by the thickness or unevenness of the diffusion layer and the phase change.
The answer on the question is in the paragraph.
Comparing the schemes of the potentiodynamic curves of bimetals after rolling in the grooves, presented in Fig. 11, it can be seen that their schemes are impacted by both the rolling method and the rolling temperature. As can be seen, the bimetals after rolling in classic grooves (variant I) are marked by the lowest values of the corrosion potential (Ecorr), and thus the earliest etching of the surface. In addition, the highest values of anode currents (ia) are also found for this type of rolling. Modification of the rolling process (variant II) caused a significant shift of Ecorr towards more positive values. Such an in-crease of Ecorr is observed for samples after rolling according to variant II (system of modified passes), regardless of the temperature used in this process, and it proves that the processes of active etching of the surface are delayed.
On the last part of question we fully agree with the proofreader’s comment that the analysis of the effect of the thickness and unevenness of the diffusion layer and the phase change on the corrosion resistance would be a valuable contribution. Nevertheless, we would like to assure you that we are planning to write another publication on the subject matter with a new structural analysis, which would include a more comprehensive set of tests, including EBSD analysis.
- The author claims that the ia of AZ31 alloy reach values of approx. 350 mAcm-2 in line 344. However, the value is 270 from Table 4. Some other comments have the same problems. Please check the data.
All values were checked and modified
- The writing of unit should be standardized. The ‘mAcm-2’ in Fig.12 and in whole paper should be uniform with the Table 4.
All units were changed according to standards
- The reason why and how the microstructure is influenced by the shape of the passes should be further discussed.
We also fully agree that the explanation of the effect of the passes on the microstructure would be a valuable contribution. Nevertheless, we would like to assure you that we are planning to write another publication on the subject matter with a new structural analysis, which would include a more comprehensive set of tests, including EBSD analysis.

Reviewer 2 Report
This manuscript presented the results of experimental tests of the rolling process of Mg/Al bimetallic bars in two systems of classic and modified passes at two temperatures of 300 and 400°C. The authors found that the use of modified passes and a lower rolling temperature resulted in more homogenous distribution of the plating layer and even grain decreasing, which improved the corrosion resistance of bimetallic bars. It was well written, and the conclusions were convincing, though the analysis was not deep enough. Some minor revision is suggested.
- In Fig. 3b), pass No. 3, duplicate “R8” should be deleted.
- The meaning of the sentence in Lines 208-110 was not clear, and should be clarified.
- In Line 301, “The most damaged was a part…” corresponding to which part, which figure?
- The photo quality of Fig. 9 was not satisfactory.
- In Line 342, “times” seems should be substituted by “orders”.
- In Line 345, what is the meaning of “4÷8”?
- References 43, 44 and 48 should be translated into English.
Author Response
REVIEWER #2
The additional text was marked as yellow.
- In Fig. 3b), pass No. 3, duplicate “R8” should be deleted.
Fig. 3(b) was changed
- The meaning of the sentence in Lines 208-110 was not clear, and should be clarified.
The sentences were modified.
According to the data in Fig. 6 the lower rolling temperature resulted in a smaller spreading and greater elongation of the Mg/Al band. It influences on the limitation of the uncontrolled flow down of the Al layer in individual passes, thanks to which the average thickness of the plating layer is greater compared to bars rolled at the temperature of 400°C where we can observe higher flowing down of the cladding layer.
- In Line 301, “The most damaged was a part…” corresponding to which part, which figure?
The authors agree that there was no information on which figure corresponds to the sentence. The authors reconstructed the sentence.
The most damaged part of the material passes modified at the temperature of 300°C (Fig. 9b) is the central part of the 1050A alloy layer, where there are areas characterized by relatively small grain size (31,4÷5,0 µm).
- The photo quality of Fig. 9 was not satisfactory.
The authors changed the Fig.9 on new, clearer with the more visible grain boundaries
|
|
||||
|
(a) |
(b) |
||||
|
|
||||
|
(c) |
(d) |
Figure 9. The 1050A microstructure after rolling in classic passes (variant I), temperature of 300°C (a); modified passes (variant II), temperature of 300°C (b); classic passes (variant I), temperature of 400°C (c); modified passes (variant II), temperature of 400°C (d).
- In Line 342, “times” seems should be substituted by “orders”.
In our opinion “times” is proper. It means that in the applied corrosive environment, aluminum anode currents has 3 times higher values than the magnesium alloy
- In Line 345, what is the meaning of “4÷8”?
It means a pH value. The sentence was changed: As it is known, in solutions with a pH of range from 4 to 8,…
- References 43, 44 and 48 should be translated into English.
Literature was translated into English
- Bialobrzeski, A.; Czekaj, E.; Heller, M. Corrosive properties of aluminum and magnesium alloys processed by pressure casting technology. Arch. Foundry 2002, 2, 294-313.
- Pourbaix, M. Lectures on electrochemical corrosion, PWN: Warszawa, Poland, 1978.
- Ura-Bińczyk, E.; Bałkowiec, A.Z.; Mikołajczyk, Ł.; Lewandowska, M.; Kurzydłowski, K.J. The influence of grain size on the corrosion resistance of the 7475 aluminum alloy. Ochr. Koroz. 2011, 2, 44-47.

Reviewer 3 Report
The manuscript “Effect of the shape of rolling passes and the temperature on the corrosion protection of the Mg/Al bimetallic bars” is dedicated to the development of new rolling methods. The aim of the work was to determine the impact of the shape of the grooves and the rolling temperature on the corrosion resistance of the bars and the authors have succeeded in achieving this. In this work was concluded that the use of modified passes and a lower rolling temperature (300°C) ensures a more homogenous distribution of the plating layer around the circumference of the core and results in an even grain decreasing, which improves the corrosion resistance of bimetallic bars compared to rolling bars in a classic system of passes and at a higher temperature (400°C). The results are reliable. I appreciate the paper design and suggest that the paper can be published in present form.
Author Response
REVIEWER #3
Dear Reviewer,
Thank you again for your review about our manuscript

Round 2
Reviewer 1 Report
In this paper, designed elongating passes and rolling temperature were investigated to improve the corrosion resistance of Mg/Al bimetallic bars. According to the results, the improvement of corrosion resistance of Mg/Al bimetallic bars is due to grain fragmentation (grain refinement) and uniform deformation of Al layers. However, the data and analysis provided by the author are not satisfactory and sufficient. Some important aspects must to be improved to match the quality of this journal.
(1) The microstructure of the alloy determines the properties of the alloy, including corrosion resistance. Therefore, the characterization of microstructure is particularly important. It is difficult for the readers to see the change of grain size from the Figure 9. EBSD or anode film may be useful.
(2) For Al layers, the reason why grain refinement is conducive to the improvement of corrosion resistance should be explained in detail.
(3) The corrosion resistance of Al layer is related not only to grain refinement, but also to texture and dislocation density. In addition, the corrosion resistance of the Mg/Al bimetallic bars should be analyzed by observing the surface corrosion morphology, and the diffusion layer formed at the Al/Mg interface also has a great influence on the corrosion resistance of Mg/Al bimetallic bars, especially long-term corrosion resistance.
(4) In addition to potentiodynamic polarization curves, electrochemical impedance spectroscopy is a good method to analyze the corrosion resistance of protective layers.
(5) The reason why the microstructure is influenced by the shape of the passes should be further discussed.
(6) There is no discussion section in this paper.
(7) The marks in Figure 12 are not well distinguished.
Author Response
Dear Reviewer,
Thank you for your thoughtful comments and helpful suggestions about our manuscript. We have read your comments carefully and detailed modifications have been made accordingly. All the changes are highlighted in yellow in the revised version.
Yours sincerely,
Sebastian Mróz
REVIEWER #1
The additional text was marked as yellow.
- The microstructure of the alloy determines the properties of the alloy, including corrosion resistance. Therefore, the characterization of microstructure is particularly important. It is difficult for the readers to see the change of grain size from the Figure 9. EBSD or anode film may be useful.
The authors fully agree with the comment that from Figure 9 it is difficult to see the change of grain size. In this case, the authors decided to change Fig.9 on new, obtained with higher magnification than previously. EBSD or anode film technique is recommended to get a more valuable microstructure observation and description but, currently, the authors laboratories are not equipped with EBSD detector but for the next publications, the authors will perform EBSD measurements in cooperation with other familiar research units.
|
|
||||
|
(a) |
(b) |
||||
|
|
||||
|
(c) |
(d) |
Figure 9. The 1050A microstructure after rolling in classic passes (variant I), temperature of 300°C (a); modified passes (variant II), temperature of 300°C (b); classic passes (variant I), temperature of 400°C (c); modified passes (variant II), temperature of 400°C (d).
2 For Al layers, the reason why grain refinement is conducive to the improvement of corrosion resistance should be explained in detail.
Included in the text
Similar results of an increase in the corrosion resistance of aluminum as a result of grain refinement were obtained after the forging (RS) [47] and angular pressing (ECAP) [48] processes. The authors attributed the increase in the corrosion resistance of the material to the formation of a compact and tight passive layer, the formation of which was fostered by both the high density of grain boundaries and dislocations. Detailed considerations on the influence of grain refinement on the corrosion resistance of aluminum are presented in [49]. While confirming the beneficial effect of grain grinding on the corrosion resistance of Al, the authors point out that the intensity of this influence depends on the corrosive environment. They also note that both the increase in the dislocation density and the associated stresses as well as the texture may affect the corrosion rate.
3 The corrosion resistance of Al layer is related not only to grain refinement, but also to texture and dislocation density. In addition, the corrosion resistance of the Mg/Al bimetallic bars should be analyzed by observing the surface corrosion morphology, and the diffusion layer formed at the Al/Mg interface also has a great influence on the corrosion resistance of Mg/Al bimetallic bars, especially long-term corrosion resistance.
The authors agree that the corrosion resistance of metallic materials depends on their structure. It should take into account not only grain refinement, but also the texture and density of dislocation as well as any irregularities in the structure. In the presented work, the authors focused on determining the corrosion resistance of only the surface of the bimetal obtained after various variants of rolling. The combination of corrosion resistance with the internal structure of bimetallic rods will be the subject of another work. Currently, research is underway on the in-depth analysis of the corrosion resistance on the cross-section of the Mg / Al bars. To assess changes in corrosion resistance, a specially developed and successfully used by the authors of the so-called the method of progressive thinning [1-3].
- K. Jagielska-Wiaderek, Depth Corrosion Characteristics of Borided Layer Produced on AISI 321 Stainless Steel, Acta Physica Polonica A, Vol. 135, 252-255 (2019) DOI: 10.12693/APhysPolA.135.252
- K. Jagielska-Wiaderek , The structure, properties and change in the cross-sectional corrosion resistance of a nitrided layer produced on AISI 321 steel, Int. J. Surface Science and Engineering, Vol. 10, No. 5,503-513 ( 2016)
- K. Jagielska-Wiaderek H. Bala, T. Wierzchoń, Corrosion depth profiles of nitrided titanium alloy in acidified sulphate solution, Central European Journal of Chemistry, 11(12), 2005-2011 (2013)
4 In addition to potentiodynamic polarization curves, electrochemical impedance spectroscopy is a good method to analyze the corrosion resistance of protective layers.
The authors fully agree with the comment of the reviewer regarding the possibility of using impedance spectroscopy in corrosion studies of surface layers. In this study, the focus was on plotting potentiodynamic polarization curves in order to assess the corrosion resistance of Mg / Al bimetals.
In subsequent corrosion tests, this type of research technique will probably also be used
5 The reason why the microstructure is influenced by the shape of the passes should be further discussed.
The additional text was added:
The implemented modification of the passes, consisting in substituting the single-radial pass surface with multi-radial surfaces with straight line segments, had the favourable effect of changing the strain, stress distribution and the plastic flow of individual components in the Mg/Al bimetallic bar in the roll gap during rolling. It effects on the higher decreasing of the grain size. The modified roll pass design helped to limit the build-up of the cladding layer material and the ends of the passes in the direction of the axis of symmetry of the pass and, at the same time, to reduce the layer thinning in the rolling reduction direction. The new pass shape of passes reduces the non-uniform distribution of the deformation and stress value In the rolling direction. Also, lowering the rolling temperature to 300°C had a favourable effect on the homogeneity of the deformation and plastic flow of the bimetal components and consequently the decreasing of the grain size. The most uniform distribution of the cladding layer and lower grain size was obtained for the modified passes for the rolling temperature of 300°C.
6 There is no discussion section in this paper.
The discussion paragraph was added into chapter Results
7 The marks in Figure 12 are not well distinguished.
The readability of Figure 12 has been improved.
